# Controlling enzyme hydrolysis of branched polymers synthesised using transfer-dominated branching radical telomerisation *via* telogen and taxogen selection
Samuel Mckeating[1], Oliver B. Penrhyn-Lowe[1], Sean Flynn[1], Savannah R. Cassin[1], Sarah Lomas[1], Christopher Fidge[2], Paul Price[2], Stephen Wright[1], Pierre Chambon[1] & Steve P. Rannard [1] ✉

With the ever-growing reliance on polymeric materials for numerous applications, new avenues to induce, design and control degradation are clearly important. Here, we describe a previously unreported approach to controlling enzymatic hydrolysis of high molecular weight branched polymers formed from the new free-radical polymer synthesis strategy transfer-dominated branching radical telomerisation (TBRT). Modifying the chemical nature of TBRT polymers may be accomplished through telogen selection and multi-vinyl taxogen (MVT) design, and we show telogen-driven control of enzyme-catalysed hydrolysis and the impact of careful placement of hydrolytically susceptible groups within readily synthesised MVTs. Our results indicate that utilising conventional free-radical chemistries and unsaturated monomers as feedstocks for highly branched polymer architectures has considerable potential for the design of future materials that degrade into very low molecular weight byproducts at variable and controllable rates.

Modern life requires polymeric materials to fulfil a range of essential functions within numerous technologies including hand-held electronics, clothing, transport, consumer products and energy generation. Across many applications, durability is a key property, however, this is also a contributor to many of the undesirable effects of plastic/polymer waste. This is particularly of concern in the absence of effective recycling processes and unintended release into the environment. In densely populated low- and middle-income nations, municipal waste management infrastructure is often below global benchmarks[1]. Waste management may also include intentional disposal within the environment through controlled land-fill sites, but polymer degradation can be compromised by anaerobic conditions and a lack of ultraviolet light[2]. The definition of a biodegradable polymer varies between sources, but a consensus has been agreed that a critical feature is the metabolic action of naturally occurring microorganisms leading to a breakdown of the substrate and the formation of $CO_2$, water, and, potentially, biomass[3].

A number of different breakdown mechanisms have been included within the scope of so-called 'degradable polymers', including the use of chemical[4,5], photolytic[6,7], and thermal[8] stimuli to trigger chemical bond scission. In many cases, placing a small number of susceptible bonds within larger macromolecular structures causes fragmentation but does not achieve wholesale degradation into small molecules[4–8]. For the majority of commercially available polymers generated by conventional chain-growth chemistries, extended backbones containing C-C bonds pose specific issues for environmental degradation and a relatively small number of cleavable groups or crosslinks does not mitigate the longer-term accumulation of non-degradable fragments with appreciable molecular weight. Alternating copolymerisation approaches have been utilised to introduce cleavable units into chain-growth polymers and avoid long segments derived from C-C bond formation[9]. Additionally, homopolymerisation of monomers such as cyclic ketene acetals *via* radical ring-opening polymerisation generate polymers with repeating ester backbones[10].

[1]Department of Chemistry & Materials Innovation Factory, University of Liverpool, Crown Street, Liverpool, L69 7ZD, UK. [2]Unilever R&D, Port Sunlight Laboratory, Quarry Road East, Bebington, Wirral, CH63, 3JW, UK. ✉e-mail: srannard@liverpool.ac.uk

Branched polymers are a specific polymer class that is ubiquitous in academic research and industrial products. Commercial examples are formed using both step-growth chemistries (eg Hybrane™ and Boltorn™) and chain-growth/ring-opening polymerisation (eg Carbomer® and Lupasol®), and they enable markets from paper production, car refinishing coatings, laundry detergents, and gene transfection[11,12]. Branched polymers from step-growth routes are generally relatively low molecular weight materials (weight average molecular weight $M_w < 20,000$ g mol$^{-1}$) and those formed from chain-growth/ring-opening chemistries are typically high molecular weight ($M_w > 100,000$ g mol$^{-1}$) non-degradable polymers[9,10]. To the best of our knowledge, there are no current commercial polymers resulting from free radical ring opening polymerisation of cyclic ketene acetals[13]; however, recent studies have shown that relatively light branching within these materials, resulting from uncontrolled back-biting reactions and initiation from the polymer backbone, may impede hydrolysis and biodegradation[10].

We recently reported, a new synthetic strategy for high molecular weight branched polymers that employs conventional free radical chemistries to form materials with backbones that are analogous to step-growth polymers[14]. Transfer-dominated Branching Radical Telomerisation (TBRT)

employs telomerisation conditions to control the homopolymerisation of polyunsaturated substrates, resulting in complete consumption of vinyl group functionality and the avoidance of gelation, Fig. 1. The term *telomerisation* was introduced over 70 years ago, with a *telomer* being the product of the addition of a *telogen* to an unsaturated *taxogen* with limited propagation[15,16]. Within a telomer distribution structures will be formed from limited propagation, including addition of a telogen to a single taxogen C = C bond, leading to number average degrees of polymerisation ($DP_n$) of <5 taxogen units[17,18]. In recent years, the focus for telomerisation has been the formation of increasingly efficient catalytic processes for the conversion of unsaturated molecules, including renewable feedstocks, to a range of functional raw materials for the chemical industry[19,20].

TBRT utilises multi-vinyl taxogens (MVTs), such as dimethacrylates, and limits the overall propagation to a $DP_n < 2$ taxogen units under ideal conditions; the formation of a large number of $DP_1$ sub-units is critical to the avoidance of network formation[21,22]. By preventing extensive C-C backbone formation, TBRT generates branched macromolecules, Fig. 1A, that are predominantly comprised of extended chains formed by the linking chemistry of the selected MVT, Fig. 1B. Simply put, using dimethacrylate MVTs generates high molecular weight branched polyesters and other

**Fig. 1 | Transfer-dominated branching radical polymerisation (TBRT) of multi-vinyl taxogens.**
**A** Schematic representation of the use of a small molecule telogen to control the homopolymerisation of a multi-vinyl taxogen by ensuring the formation of telomer subunits that control the propagation to an average of <2 vinyl units.
**B** Structure of a TBRT polymer synthesised from a dimethacrylate multi-vinyl taxogen and utilising a thiol telogen; the extended backbone of the polymer (shown in yellow) is a polyester, derived from the ester units within the multi-vinyl taxogen.

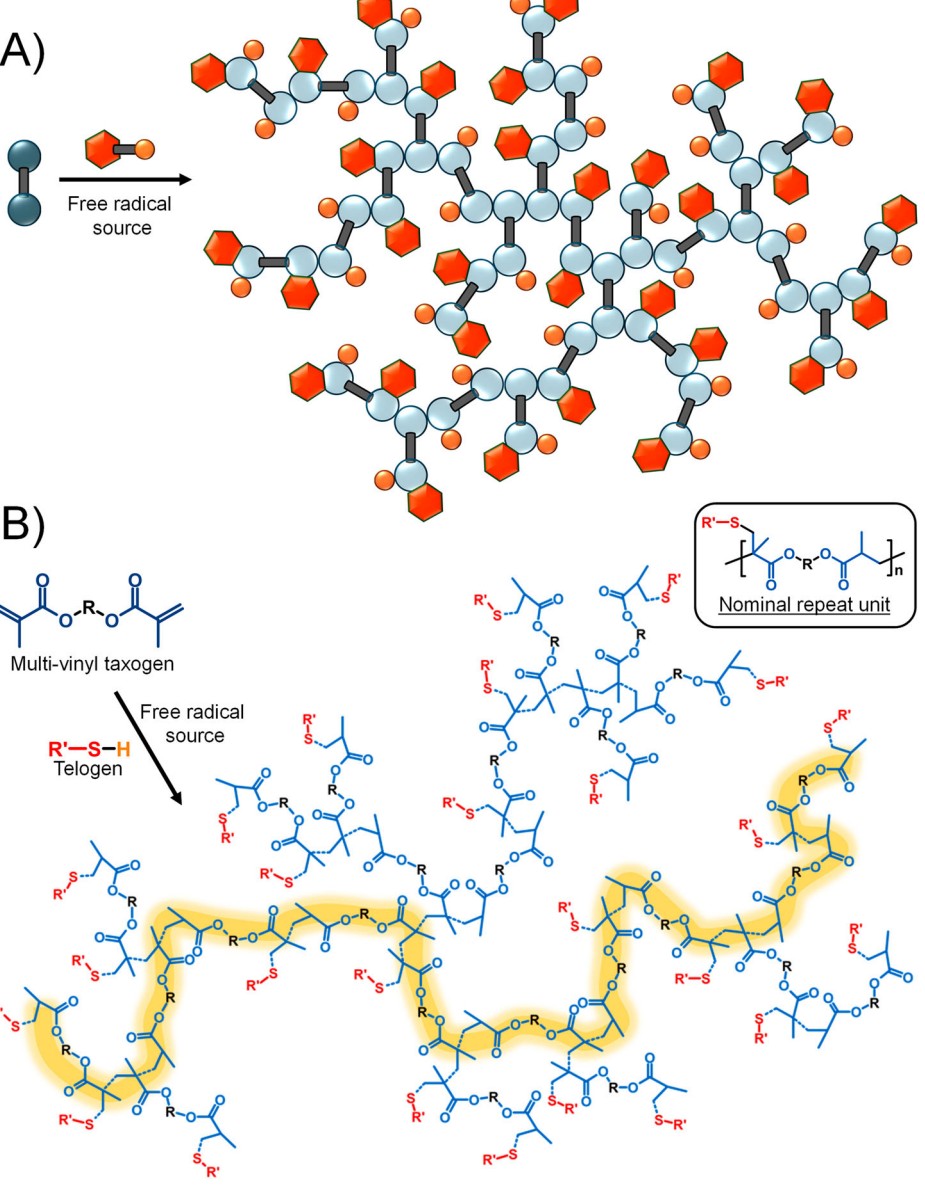

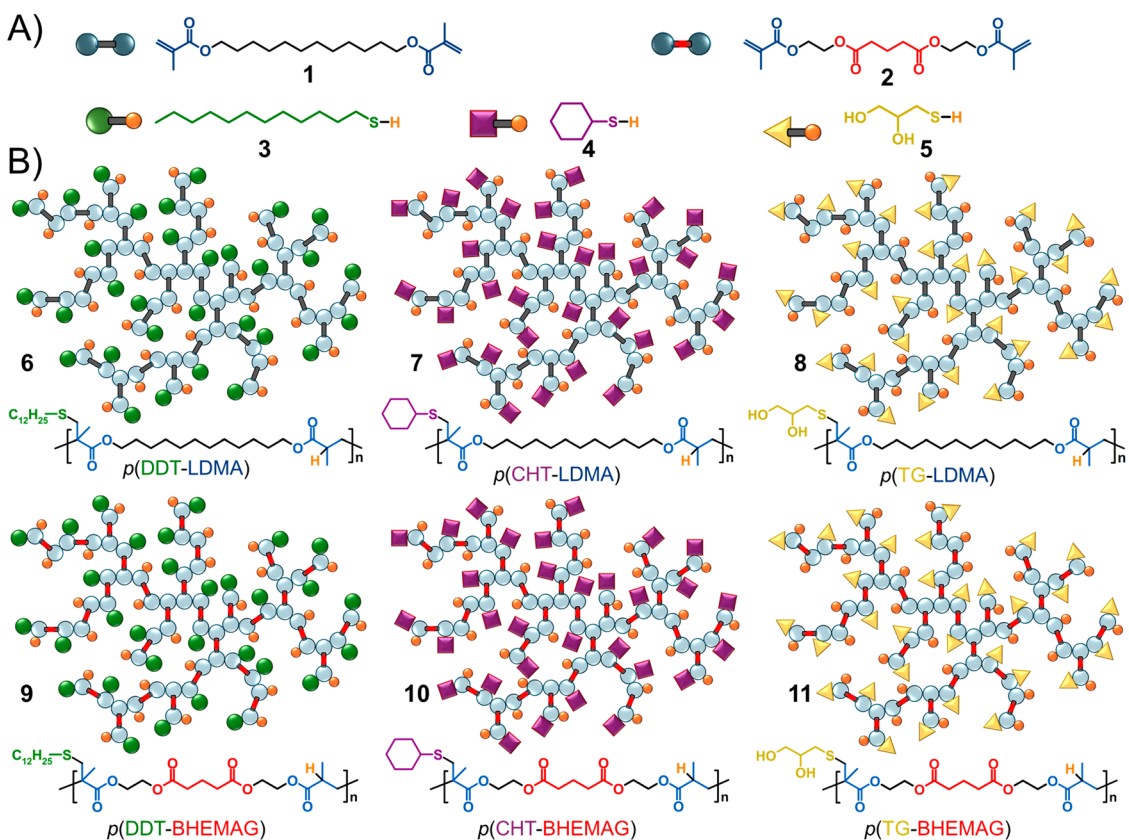

**Fig. 2 | Starting materials and resulting TBRT polymers. A** Multi-vinyl taxogens: lauryl dimethacrylate, **1**, and bis-HEMA glutarate, **2**. Telogens: 1-dodecanethiol, **3**, cyclohexanethiol, **4**, and 1-thioglycerol, **5**. **B\** Resulting branched polymers derived from lauryl dimethacrylate polymerisation with each telogen (**6**, **7**, and **8**), and the polymerisation of bis-HEMA glutarate with each telogen (**9**, **10**, and **11**).

backbones may be selected through the appropriate choice of MVT[23]. Other unique features of TBRT include the controlled introduction of cyclisation within the branched polymers when varying solids concentrations[24], and the introduction of telogen in a 1:1 molar ratio with the chosen MVT, thereby introducing a pendant chain to the polyester backbone within the nominal repeat unit *during* polymerisation, Fig. 1B[25]. Pendant chain variation is simply modified by selecting a different telogen, leading to multiple options for entirely novel polymer structures[26].

Here, we demonstrate the ability to impart enzymatic hydrolysis into high molecular weight branched polymers formed by TBRT and control the rate of degradation. Unexpectedly, a significant impact of telogen chemistry was observed, providing an entirely new telogen strategy for enabling and altering the rate of polymer degradation, to the best of our knowledge. The synthesis and use of a novel MVT was also studied to create a taxogen-strategy for introducing degradability into these complex macromolecules. Novel mixed telogen and mixed MVT copolymers have been shown to allow fine tuning of degradation, providing considerable opportunities for future material design. Finally, a detailed study of the water-soluble degradation products from the TBRT polymers has shown the formation of telomer fragment distributions with no detectable molecular weight > 1600 g mol⁻¹, supporting the premise that, despite being synthesised using free-radical C-C bond forming reactions, TBRT polymers may be designed to degrade to very low molecular weight fragments under mild conditions.

## Results
### Telogen and taxogen control of enzymatic degradation of polymers formed by TBRT

The TBRT of dimethacrylate monomers yields a branched polyester backbone with pendant telogen-derived groups, Fig. 1B. The enzymatic degradation of polyester backbones is not, however, a given feature of

such polymers; substitution neighbouring ester carbonyls may yield poor degradation behaviour in the presence of water-borne enzymes, especially for hydrophobic polymers. In recent years, the isolation of new enzymes capable of degrading $p$(ethylene terephthalate), PET, from marine micro-organisms has opened new avenues of investigation for bioremediation of waste[27,28]; however, we selected to study the action of esterase, lipase and protease enzymes that are relatively commonly used in industrial products or processes.

Lauryl dimethacrylate (LDMA, **1**) is a commercially available bifunctional monomer that has been employed previously as an MVT in TBRT studies[22]. The synthesis of three TBRT homopolymers containing the LDMA backbone was readily accomplished using either 1-dodecanethiol (DDT, **3**), cyclohexanethiol (CHT, **4**), or 1-thioglycerol (TG, **5**), Fig. 2A, leading to high molecular weight polymers varying only in their side chain chemistry, Fig. 2B (Supplementary Figs. S1-13; Table S1). The synthesis of the three polymers, $p$(DDT-LDMA), $p$(CHT-LDMA), and $p$(TG-LDMA) was studied using varying [MVT]$_0$/[Telogen]$_0$ molar ratios within the reactions and optimised to provide samples with similar molecular weights and dispersities ($Đ$). As seen previously, high molecular weight and soluble $p$(DDT-LDMA) was synthesised at an [MVT]$_0$/[DDT]$_0$ molar ratio of 0.54, yielding samples with a number average molecular weight ($M_n$) of 19,480 g mol⁻¹ and a weight average molecular weight ($M_w$) of 249,900 g mol⁻¹ using 2,2'-azobis(isobutyronitrile) (AIBN) as the free radical source and 50 wt% solids concentration in ethyl acetate at 70 °C. Due to the relatively poor chain transfer from the secondary thiol of CHT, the highest [MVT]$_0$/[Telogen]$_0$ molar ratio achieved yielding soluble polymer was 0.34 ($p$(CHT-LDMA): $M_n$ = 10,490 g mol⁻¹; $M_w$ = 561,090 g mol⁻¹); similarly, reactions with [MVT]$_0$/[Telogen]$_0$ molar ratios >0.37 were unable to control the TBRT reaction when using TG, but soluble $p$(TG-LDMA) homopolymer was obtainable at this ratio ($M_n$ = 21,220 g mol⁻¹; $M_w$ = 893,850 g mol⁻¹).

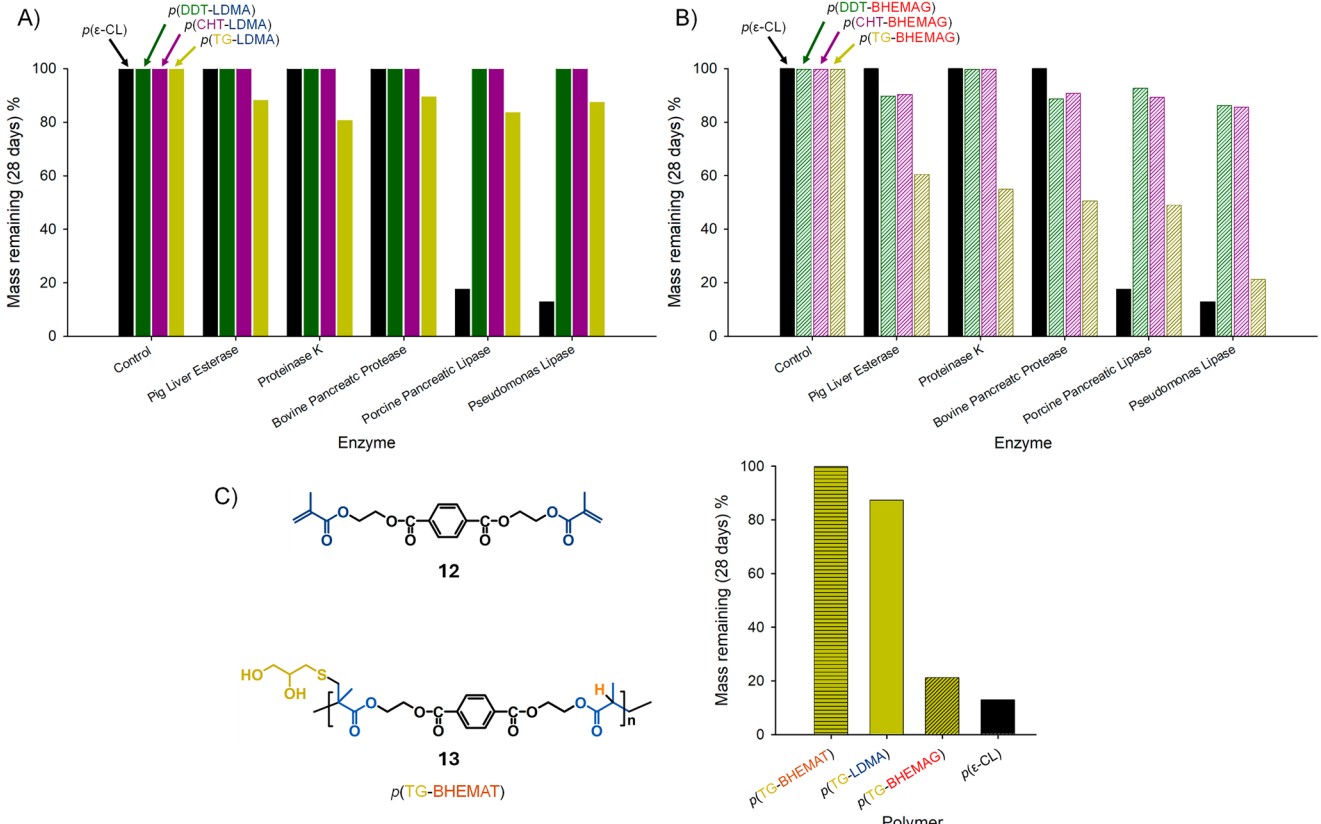

**Fig. 3 | Evaluation of enzymatic hydrolysis of TBRT polymers using five enzymes ($p$(ε-caprolactone) as linear polymer standard). A** Comparative hydrolysis of polymers synthesised using lauryl dimethacrylate; **B** Comparative hydrolysis of polymers synthesised using bis-HEMA glutarate; **C** Structure of bis-HEMA terephthalate, **12**, branched polymer resulting from the TBRT of bis-HEMA terephthalate with 1-thioglycerol, **13**, and comparison of mass loss *via* enzymatic hydrolysis of 1-thioglycerol containing TBRT polymers when exposed to and *pseudomonas* lipase for 28 days.

In all cases, complete consumption of vinyl groups was observed within the ${}^1$H nuclear magnetic resonance (NMR) spectra of the crude reaction mixtures. As we have reported, the [MVT]$_F$/[Telogen]$_F$ ratio within the purified polymers is highly indictive of the presence of cycles within these branched polymers[24], but each of these polymers showed values close to unity, indicating a near ideal branched structure ($p$(DDT-LDMA) = 1.01; $p$(CHT-LDMA) = 1.06; $p$(TG-LDMA) = 1.04). The polymers were hydrophobic, fully soluble in ethyl acetate, and exhibited glass transition temperatures ($T_g$) which varied with telogen, as seen in our previous reports of TBRT materials ($p$(TG-LDMA = -67 °C; $p$(DDT-LDMA) = –24 °C; $p$(CHT-LDMA) = –6 °C).

An initial degradation screen (Supplementary Figs. S14-15, Equation S1) of the three LDMA-derived polymers utilised pig liver esterase, proteinase K, bovine pancreatic protease, porcine pancreatic lipase, and *Pseudomonas* lipase; the absence of enzyme was employed as a negative control and a $p$(ε-caprolactone) sample chosen as a positive control ($p$(ε-CL)). Mass loss was used to establish the extent of enzyme catalysed hydrolysis and indicate susceptibility for degradation. Each polymer was placed in a vial with an aqueous solution of enzyme in phosphate buffered saline (PBS; pH = 7.2) and incubated for a period of 4 weeks at 37 °C with each enzyme being regularly replenished during the study. For the $p$(ε-CL) control polymer, there was no discernible difference in polymer degradation in the absence of enzyme and when using pig liver esterase, proteinase K, and bovine pancreatic protease, with no mass loss observed during this study. This was also seen for all enzyme and control conditions when studying both $p$(DDT-LDMA) and $p$(CHT-LDMA), Fig. 3A.

Significant mass loss was observed when $p$(ε-CL) was treated with porcine pancreatic lipase and *Pseudomonas* lipase (Supplementary Fig. S15), but surprisingly, $p$(TG-LDMA) showed susceptibility to hydrolysis across all enzymes studied under these conditions, ranging from 10.7–19.5% mass

loss. Given that the structural variation of these TBRT polymers is purely derived from telogen (pendant group), variation of the susceptibility towards degradation by the broad range of enzymes appears to be telogen driven, presenting a new structural feature to induce degradation within high molecular weight polymers, Fig. 3A.

To study the impact of MVT chemistry on enzyme degradation, a novel dimethacrylate MVT was synthesised by coupling two 2-hydroxethyl methacrylate (HEMA) monomers using glutaryl chloride to form bis-HEMA glutarate (BHEMAG, **2**), Fig. 2A (Supplementary Fig. S16–18; Table S2). LDMA and BHEMAG have just a single atom difference in chain length and were therefore expected to show similar reactivity under TBRT conditions[22]. The introduction of two unhindered esters within the MVT, and ultimately the nominal repeat unit within the resulting polymers, Fig. 2B, was expected to enable enzymatic degradation. This MVT was subjected to TBRT using the same conditions employed for LDMA and the same series of DDT, CHT and TG telogens (Supplementary Figs. S19–30; Table S3). The outcomes were remarkably similar, with soluble branched homopolymers synthesised at [MVT]$_0$/[Telogen]$_0$ molar ratios of 0.35 using CHT ($p$(CHT-BHEMAG): $M_n$ = 12,200 g mol⁻¹ ; $M_w$ = 91,700 g mol⁻¹), 0.40 in the presence of TG ($p$(TG-BHEMAG): $M_n$ = 14,200 g mol⁻¹; $M_w$ = 489,400 g mol⁻¹), and 0.55 for reactions containing DDT ($p$(DDT-BHEMAG): $M_n$ = 19,620 g mol⁻¹; $M_w$ = 234,100 g mol⁻¹). Values for $T_g$ were established as –58 °C, –42 °C and -4 °C for $p$(TG-BHEMAG), $p$(DDT-BHEMAG), and $p$(CHT-BHEMAG), respectively, following the same telogen-derived trend as the LDMA-derived TBRT polymers. The [MVT]$_F$/[Telogen]$_F$ molar ratios within the final purified polymer samples were determined as $p$(DDT-BHEMAG) = 1.03, $p$(CHT-BHEMAG) = 1.01, and $p$(TG-BHEMAG) = 1.07, also falling within a very similar range as the polymers employing LDMA.

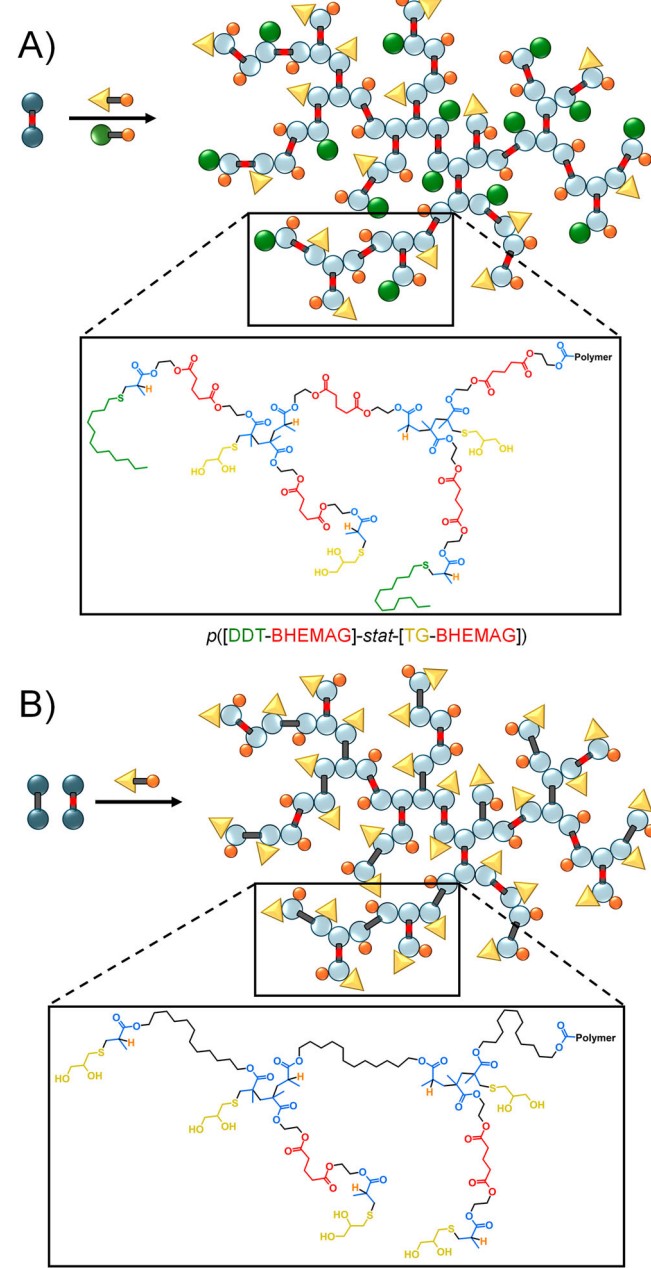

p([DDT-BHEMAG]-stat-[TG-BHEMAG])

p([TG-LDMA]-stat-[TG-BHEMAG])

**Fig. 4 | Schematic representation of the synthesis of TBRT copolymer structures.** **A** Mixed telogen copolymer and structure of resulting polymer derived from the TBRT of bis-HEMA glutarate with a 1:1 mixture of 1-dodecanethiol and 1-thioglycerol; B) Mixed multi-vinyl taxogen copolymer and structure of resulting polymer derived from the TBRT of a 1:1 mixture of lauryl dimethacrylate and bis-HEMA glutarate with 1-thioglycerol.

When subjected to the range of enzyme degradation studies described above, all of the BHEMAG-derived polymers showed considerably enhanced susceptibility to hydrolysis, Fig. 3B (Supplementary Fig. S31). *p*(DDT-BHEMAG) and *p*(CHT-BHEMAG) exhibited very similar rates of enzyme catalyzed hydrolysis across all enzymes, with mass losses ranging from 7.3–14.4%, except when proteinase K was used. Importantly, *p*(TG-BHEMAG) again displayed the telogen-directed enhanced susceptibility to enzymatic cleavage, leading to considerable mass losses (39.5–78.8%) across all enzymes. These results indicate that the branched TBRT polymers derived from the BHEMAG MVT are more widely susceptible to different enzymes than *p*(ε-CL), with *p*(TG-BHEMAG) outperforming this

established degradable polymer in the presence of pig liver esterase, proteinase K, and bovine pancreatic protease, and rivalling degradation when subjected to *Pseudomonas* lipase under these conditions.

Telogen (pendant group)-mediated enzymatic degradation has not been previously reported for branched polyesters and the impact of TG may be interpreted as enabling hydration of the polymer and, therefore, providing enzyme access to ester functional groups. Nevertheless, when BHEMAG is used within the backbone of the TBRT polymer, access is clearly still available in the presence of CHT and DDT, both of which are hydrophobic and appear to limit, but not fully prevent, the ability of each enzyme to reach the ester degradation sites (with the exception of proteinase K). The water-solubility of the hydrolysis products may also vary with varying telogen and higher degrees of hydrolysis may be required to yield fully soluble byproducts.

The reason for the enhanced susceptivity of the backbone esters for these TBRT polymers, when compared to *p*(ε-CL), is not entirely clear. Although the physical form of the polymer samples may play a role (*p*(ε-CL): film; TBRT polymers: low $T_g$ semi-solid), it is unlikely that the difference is solely based on this given the comparative differences within the series of TBRT materials. It is also worth emphasising that cleavage of the more hindered backbone esters derived purely from the methacrylate functionality is occurring within *p*(TG-LDMA), as no additional ester groups are present within the polymer backbone. To study the impact of ester chemistry in more detail, HEMA was reacted with terephthaloyl chloride to form bis-HEMA terephthalate (BHEMAT; **12**), Fig. 3C, which, again, closely resembled the molecular dimensions of LDMA and BHEMAG (Supplementary Fig. S32-33; Table S4). TBRT polymerisation of BHEMAT with TG yielded soluble branched polymers up to a [MVT]_0/[Telogen]_0 molar ratio of 0.42 (*p*(TG-BHEMAT); **13**: $M_n = 18,600$ g mol$^{-1}$; $M_w = 236,900$ g mol$^{-1}$), and after purification the [MVT]_f/[Telogen]_f molar ratio within the final polymer was measured as 1.00, indicating an ideal branched structure with few, if any, cyclic substructures (Supplementary Fig. S34–36; Table S5).

When subjected to *Pseudomonas* lipase under the same conditions described above, it was clear that *p*(TG-BHEMAT) showed no discernible mass loss over the 28-day period, Fig. 3C. Despite the presence of the additional backbone ester functionality, *p*(TG-BHEMAT) appears to not be susceptible to enzymatic cleavage through the same mechanisms acting on the other polymer samples studied. The additional impact of the inclusion of aromatic ester groups clearly indicates that TBRT offers the potential to mediate enzyme hydrolysis through telogen and taxogen selection/design, leading to polymers that are resistant or highly susceptible to degradation under these conditions.

## Controlling enzymatic degradation through TBRT copolymer design

As mentioned above, TBRT offers additional synthetic strategies for copolymer synthesis. Through mixing telogens, for example, TBRT polymers can be generated with a statistical mixture of telogen-derived side chains that mimic statistical copolymers synthesised by chain-growth polymerisation approaches. Additionally, by mixing MVTs it is possible to generate statistical copolymers that vary in their backbone chemistry, which are analogous to copolymers synthesised using mixed A$_2$ or B$_2$ monomers using step-growth polymerisation chemistries[26].

Given the impact of telogens and MVT structures on the enzymatic cleavage of the TBRT polymers, a series of copolymer structures was synthesised in order to investigate the potential to fine tune the observed degradation profiles. Within these studies, *Pseudomonas* lipase was selected as the sole study enzyme given the variable degradation seen within the *p*(Telogen-BHEMAG) polymers and the apparent enzymatic hydrolysis susceptibility of *p*(TG-LDMA). The two telogens selected for mixed telogen copolymer synthesis were DDT and TG, whilst LDMA and BHEMAG were the obvious choices for mixed MVT copolymer synthesis, Fig. 4A.

Mixed telogen copolymers with a range of compositions were synthesised simply by mixing TG and DDT at varying concentrations of the

**Table 1 | $^1$H nuclear magnetic resonance (NMR) and triple detection size exclusion chromatography (TD-SEC) analyses of mixed telogen and mixed multi-vinyl taxogen copolymers using synthesised using TBRT**

| Statistical TBRT Copolymers (telogen-MVT) | $^1$H NMR (CDCl$_3$)[a] | | TD-SEC (THF) | | | |
|---|---|---|---|---|---|---|
| | [MVT]$_0$/[Tel]$_0$ (initial reaction) | [MVT]$_f$/[Tel]$_f$ (purified polymer) | $M_w$ (g mol$^{-1}$) | $M_n$ (g mol$^{-1}$) | Đ | MHS α |
| | Mixed Telogen Copolymers | | | | | |
| p([DDT-BHEMAG]$_{75}$-stat-[TG-BHEMAG]$_{25}$) | 0.45 | 0.94 | 106,200 | 15,100 | 7.0 | 0.341 |
| p([DDT-BHEMAG]$_{50}$-stat-[TG-BHEMAG]$_{50}$) | 0.55 | 1.03 | 636,800 | 22,400 | 28.3 | 0.349 |
| p([DDT-BHEMAG]$_{25}$-stat-[TG-BHEMAG]$_{75}$) | 0.55 | 0.97 | 54,100 | 5,670 | 9.5 | 0.338 |
| | Mixed MVT Copolymers | | | | | |
| p([TG-LDMA]$_{75}$-stat-[TG-BHEMAG]$_{25}$) | 0.36 | 0.96 | 147,900 | 8,010 | 18.4 | 0.471 |
| p([TG-LDMA]$_{50}$-stat-[TG-BHEMAG]$_{50}$) | 0.40 | 1.00 | 158,900 | 11,400 | 13.9 | 0.305 |
| p([TG-LDMA]$_{25}$-stat-[TG-BHEMAG]$_{75}$) | 0.37 | 1.05 | 725,800 | 14,100 | 51.4 | 0.439 |

[a]All polymerisations reached > 99% vinyl group consumption.

initial feedstock ratio within the TBRT reaction of BHEMAG, under the same conditions described for homopolymer synthesis (Supplementary Fig. S37, 38). To compensate for the different chain transfer coefficients of the telogens, the initial [MVT]$_0$/[Telogen]$_0$ molar ratios were varied to ensure high molecular weight soluble polymer was formed in each case with full consumption of vinyl functionality, Table 1. After purification, the mixed telogen copolymer samples varied in $M_w$ from 54,100 g mol$^{-1}$ to 636,800 g mol$^{-1}$ as the composition of the copolymers changed from p([DDT-BHEMAG]$_{25}$-stat-[TG-BHEMAG]$_{75}$) to p([DDT-BHEMAG]$_{75}$-stat-[TG-BHEMAG]$_{25}$), with no indication of additional cyclisation when compared to the homopolymer syntheses.

Similarly, when polymerising a mixed MVT feedstock of LDMA and BHEMAG using TG as the telogen under TBRT conditions, the formation of copolymers ranging from p([TG-LDMA]$_{75}$-stat-[TG-BHEMAG]$_{25}$) to p([TG-LDMA]$_{25}$-stat-[TG-BHEMAG]$_{75}$), Fig. 4B, was accomplished as previously, using [MVT]$_0$/[Telogen]$_0$ molar ratios within the range 0.36–0.40. This is almost identical to the conditions used for the synthesis of the two homopolymers p(TG-LDMA) and p(TG-BHEMAG), as might be expected (Supplementary Fig. S39-40). In our previous studies of mixed telogen and mixed taxogen branched copolymer synthesis using TBRT, there were clear indications that telogen and taxogen incorporation was relatively homogeneous throughout the branched polymer structures[26]. This is not surprising given the statistical nature of intermolecular branching reactions at high vinyl group consumption.

The enzymatic degradation conditions used previously were again employed to study the impact of *Pseudomonas* lipase on the enzyme catalysed hydrolysis of different copolymers over a 28-day incubation period but including multiple repeats ($n = 3$), Fig. 5A. Importantly, the extremes of each copolymer series are different when considering the two studies. The homopolymers for the mixed-telogen statistical copolymers are p(DDT-BHEMAG) and p(TG-BHEMAG) with variation of DDT/TG molar ratios within the systematically varying structures; the homopolymer extremes for the mixed-MVT study are p(TG-LDMA) and p(TG-BHEMAG) with LDMA/BHEMAG varying within the copolymer series.

The mixed telogen copolymers clearly demonstrated a variation in enzymatic degradation that correlates to increasing TG within the statistical copolymer composition, Fig. 5A. The inclusion of 25 mol% of TG had a relatively minor impact with a small decrease in the measured mass remaining in the sample (from 86.2% to 82.4%) after 28 days. The formation of p([DDT-BHEMAG]$_{50}$-stat-[TG-BHEMAG]$_{50}$) led to a larger decrease in mass during the experiment, to 73.1% of the starting sample, with p([DDT-BHEMAG]$_{25}$-stat-[TG-BHEMAG]$_{75}$) leading to nearly a 50% decrease over this timescale, Fig. 5A.

The inclusion of 25 mol% of BHEMAG into the mixed MVT structures was more impactful than a 25 mol% TG addition within the mixed telogen copolymers; the p([TG-LDMA]$_{75}$-stat-[TG-BHEMAG]$_{25}$) showed almost double the mass loss observed in the p(TG-LDMA) homopolymer. Further inclusion of BHEMAG into the backbone of the statistical copolymers led to

increasing mass losses. In both cases of mixed telogen and mixed MVT statistical copolymers, it is expected that the inclusion of more TG or increasing BHEMAG above the respective 75 mol% studied here, will have a dramatic impact on enzymatic degradation, as the mass remaining after degrading p(TG-BHEMAG) under these conditions is approximately 40% of these values. Fig.5A (Supplementary Tables S6–8).

The two statistical copolymer series were also subjected to a 42-day degradation study using *Pseudomonas* lipase, with samples taken at regular intervals to investigate the different hydrolysis kinetics within the two copolymer strategies, Fig. 5B, C (Supplementary Tables S6-8). Within the mixed telogen copolymer series, the p(DDT-BHEMAG) homopolymer showed an initial steady decrease in mass until 14 days, after which the sample mass stabilised, suggesting no further degradation over the following 28 days, Fig. 5B.

This may suggest that regions within the p(DDT-BHEMAG) homopolymer sample are highly accessible to the enzyme and once these have been degraded, access to the branched polymer architecture becomes increasingly difficult. The most intuitive areas that may provide accessibility are the backbone esters near the $DP_1$ terminal groups, although it is also possible that the lower molecular weight species within the distribution are responsible for this initial mass loss. The degradation profile of p([DDT-BHEMAG]$_{75}$-stat-[TG-BHEMAG]$_{25}$) appears to follow a similar trajectory to the p(DDT-BHEMAG) homopolymer but rather than a plateau, the degradation appears to continue slowly from 14 days through to the 42-day endpoint (approximately additional 9.5% mass loss during this time), Fig. 5B. The equimolar mixed telogen copolymer p([DDT-BHEMAG]$_{50}$-stat-[TG-BHEMAG]$_{50}$) displayed a much steeper decline in mass during the first 7 days of the degradation study and a similar slowing of mass loss for the remainder of the study, whilst leading to a larger overall mass loss. p([DDT-BHEMAG]$_{25}$-stat-[TG-BHEMAG]$_{75}$) led to a considerable change in degradation within the first 7 days of the study, matching the behaviour of the p(TG-BHEMAG) homopolymer. The slowing of degradation from 7 days through to the study endpoint was seen for this copolymer but was absent from the homopolymer sample, presumably due to decreasing access to the backbone as sections bearing TG side chains are cleaved, leaving DDT side chains as the predominant chemistry. The p(TG-BHEMAG) homopolymer showed steady mass loss across the 42-day study with only 14.6% of the initial sample mass remaining at the end of the study and no sign of the onset of a plateau at this point, Fig. 5B.

The mixed MVT statistical copolymers showed similar behaviour, as the LDMA within the backbone was decreased from 100 mol% through to 0 mol%. Interestingly, the degradation of p(TG-LDMA) appears to be faster than p(DDT-BHEMAG) over the first 14 days, but again a plateau was observed where no further degradation was measured over the next 28 days, Fig. 5C. Inclusion of 25 mol% BHEMAG into the statistical mixed MVT copolymer led to a noticeably faster decrease in sample mass than the inclusion of 25 mol% of TG within the mixed telogen samples (7-day

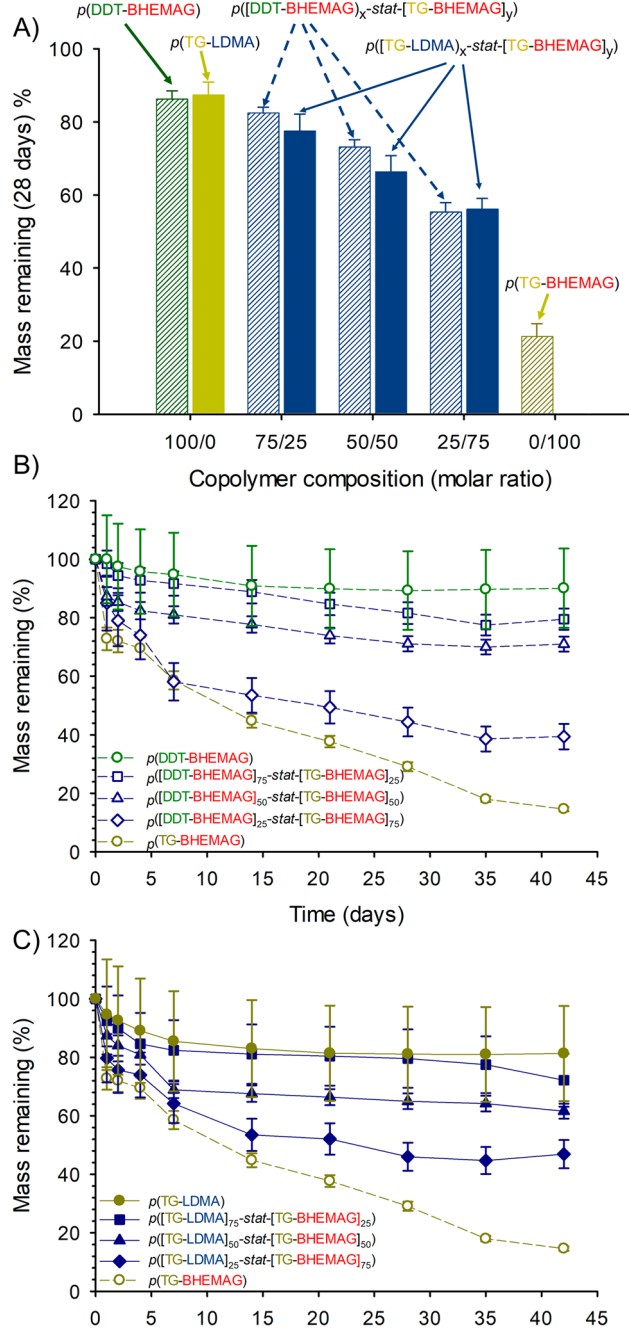

**Fig. 5 | Evaluation of enzymatic hydrolysis of TBRT copolymer series using and** ***pseudomonas* lipase. A** Comparison of weight loss for mixed telogen copolymers (solid blue bars) and mixed multi-vinyl taxogen (hashed blue bars) after 28 days exposure to enzyme. TBRT homopolymers shown for each comparative series are *p*(DDT-BHEMAG) and *p*(TG-BHEMAG) (mixed telogen copolymer series), and *p*(TG-LDMA) and *p*(TG-BHEMAG) (mixed multi-vinyl taxogen copolymer series); **B** Monitoring mass loss for the mixed telogen copolymer series over 42 days; **C** Monitoring mass loss for the mixed multi-vinyl taxogen copolymer series over 42 days.

mass loss: $p([\text{TG-LDMA}]_{75}\text{-}stat\text{-}[\text{TG-BHEMAG}]_{25}) = 17.6\%$; $p([\text{DDT-BHEMAG}]_{75}\text{-}stat\text{-}[\text{TG-BHEMAG}]_{25}) = 8.3\%$. In general, the mixed MVT statistical copolymers led to a greater mass loss than the corresponding mixed telogen samples, probably due to the hydrophilic nature of the single telogen used and the formation of soluble cleavage products even when LDMA segments had not fully degraded.

## Characterisation of enzymatic degradation products and mechanism elucidation

Identifying the byproducts of polymer degradation can be extremely challenging; however, the TBRT polymers derived from BHEMAG should preferentially undergo enzymatic cleavage at the glutaric acid residues within the polyester backbone and yield the distribution of telomers that act as structural subunits (linear, branching and terminal) within the high molecular weight branched polyester products[14,21]. When using BHEMAG, the telomer distribution should, therefore, resemble the products of a conventional telomerisation of HEMA.

To establish the conditions for characterising the theoretical degradation products, a HEMA telomerisation was carried out using TG as the telogen and under conditions similar to those used to form *p*(TG-BHE-MAG), Figure 6Ai&B (Supplementary Figs. 41–43, Table S9). Time of flight matrix assisted laser desorption ionisation (MALDI-TOF) mass spectrometry was used to study the *p*(HEMA)-TG linear telomerisation, and a distribution of ions was observed, with a clear repeating pattern of species separated by the mass of additional HEMA repeat units. Figure 6Aii.

The MALDI-TOF analysis of the isolated byproducts of the enzymatic hydrolysis of *p*(TG-BHEMAG) after exposure to *Pseudomonas* lipase, Figure 6Ci-ii, showed a distribution of species complicated by the oxidation of the thioether units within the polymer structure, Figure 6Ciii (Supplementary Figs. S44–46, Table S10). It is known that under mass spectrometry conditions, different degrees of oxidation may be seen within sulphur containing species[29,30]. Close examination of the species present clearly showed repeating patterns of low molecular weight materials separated by the HEMA repeat unit mass and equating to TG-terminated telomers with chain lengths up to 11 HEMA units ($DP_{11}$). Under these conditions, no species with a mass of > 1600 g mol⁻¹ were observed. Identification of $DP_1$ structures was accomplished using a porphyrin-based matrix and the intensity of detected telomers > $DP_4$, was significantly reduced, potentially due to ionisation differences.

## Discussion

TBRT offers access to new polymer structures and the potential to tune properties towards novel and previously unobtainable behaviour. With the ever-growing need to consider the long-term global impact of polymeric materials, this is particularly important in products that are, by necessity, intentionally released into the environment and contain formulated macromolecules. Within this study, we have shown that high molecular weight TBRT polymers can be rendered susceptible to enzymatic degradation through relatively small modification of the telogens and taxogens used in their synthesis. The enzymatic hydrolysis of *p*(TG-BHEMAG) has shown that degradation forms the linear telomer distribution that describes the subunits of the branched polymer architecture.

The telomer subunits can be considered as small molecule esters and may be compared with the expected fragments of known biodegradable polymers such as *p*(butylene glutarate), *p*(BG), and *p*(3-hydroxybutyrate-*co*-3-hydroxyvalerate), *p*(HBV). For example, the $DP_2$ telomer resulting from *p*(TG-BHEMAG) degradation, Fig. 7A, is a diol (ignoring the telogen contribution) that bears considerable similarity to the trimer structures of *p*(BG) and *p*(HBV), Fig. 7Bi&ii, which would both be formed during biodegradation of the parent polymers. It is plausible that under similar conditions, the telomers of *p*(TG-BHEMAG) would further degrade through ester cleavage to yield poly-acidic structures, Fig. 7Ci-iii. These also bear resemblance to small chains of poly(acrylic acid), Figure 7Di-iii, which are also known to biodegrade if the chains are short ($DP_n < 18$ units)[31]. The telomer distribution analysed from the degradation of *p*(TG-BHEMAG) suggested very little of the subunit distribution comprised chain lengths over 11 units (none were observed by MALDI-TOF). Full cleavage of the esters within this distribution would lead to poly(methacrylic acid) chains that may mimic the reports of degradation of short chains of poly(acrylic acid).

It is maybe a philosophical point, however, the $DP_2$ diacid telomer, Figure 7Ci, may be considered as a substituted dihydroxy glutaric acid rather than a $DP_2$ poly(methacrylic acid), and the $DP_3$ triacid telomer, Figure 7Cii,

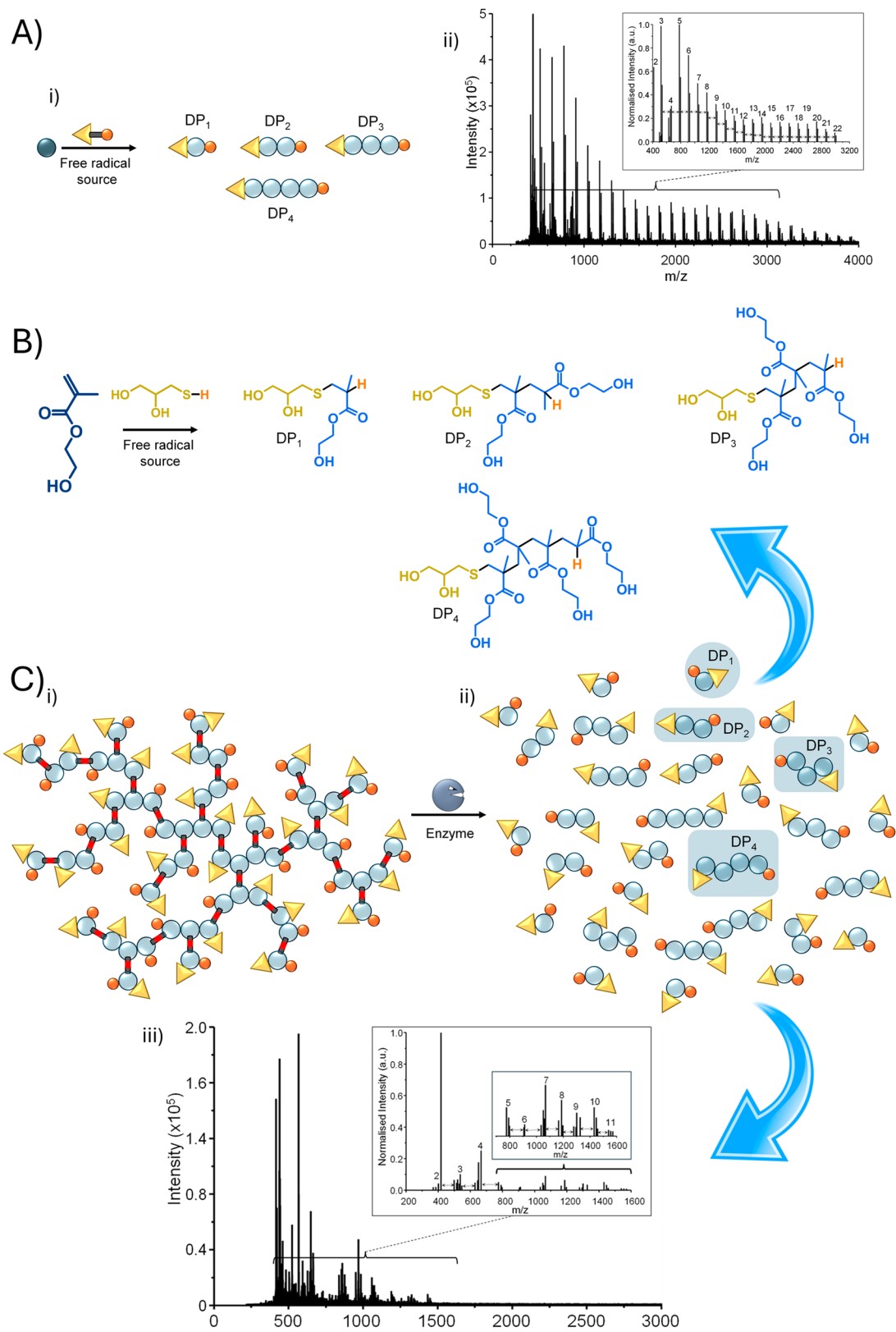

**Fig. 6 | Elucidating the mechanism of enzymatic hydrolysis. A** i) Schematic representation of the linear telomerisation of 2-hydroxyethyl methacrylate using 1-thioglycerol, and ii) the resulting MALDI-TOF mass spectrum of the telomer sample; **B** Structures of the telomers formed from the telomerisation of 2-hydroxyethyl methacrylate ($DP_{1-4}$ shown); and **C** i) Schematic representation of $p$(TG-BHEMAG), ii) the distribution of telomers that may be formed ($DP_{1-4}$ shown) and how they mirror the telomer distribution from the telomerisation of 2-hydroxyethyl methacrylate, and iii) the MALDI-TOF mass spectrum of the byproducts of enzymatic hydrolysis showing telomer substructures with masses < 1600 g mol$^{-1}$.

**Fig. 7 | Comparison of telomer byproducts after enzymatic hydrolysis with structures formed during the degradation of commercial degradable materials. A** The $DP_2$ telomer derived from the enzymatic hydrolysis of $p$(TG-BHEMAG); **B** Comparative structures formed from the degradation of i) $p$(butylene glutarate), and ii) $p$(3-hydroxybutyrate-$co$-3-hydroxyvalerate); **C** Expected structure resulting from further hydrolysis of telomers ($DP_{2-4}$ shown), yielding a $p$(methacrylic acid) telomer sample, and **D** Comparison with $p$(acrylic acid) telomers with reported biodegradability.

may be considered as a substituted dihydroxypentane-1,3,5-tricarboxylic acid. 1,3-Dihydroxypentane-1,3,5-tricarboxylic acid is a naturally occurring molecule found in the Peruvian apple cactus (*Cereus repandus*). As such, these small molecule fragments may be relatively environmentally benign and subject to further bacterial consumption.

When forming telomers by radical chemistries, the products that result are clearly derived from C-C bond formation, but where the dividing line lies between small molecules and persistent polymer chains is not completely clear and remains an area with some lacking of definition. To establish whether these materials show potential for biodegradation, studies will be required that investigate the evolution of $CO_2$ and the clear indication of metabolic activity in appropriate microbial mixtures. Work towards this aim is ongoing, with the target of showing the benign nature of the degradation products of TBRT polymers when designed with environmental concerns in mind.

## Methods
### Materials
1,12-dodecanediol dimethacrylate (LDMA, 97%) was purchased from Alfa Aesar. 2-hydroxy ethyl methacrylate (HEMA, 99%), 1-dodecanethiol (DDT, >98%), 1-cyclohexanethiol (HT, >95%), 1-thioglycerol (TG, >98%), 2,2'-azobis(2-isobutyronitrile) (AIBN, 98%), deuterated chloroform (CDCl$_3$, 99.8 atom % D), glutaryl chloride (97%), terepthaloyl chloride (97%), dichloromethane (DCM, anhydrous), triethylamine (TEA, anhydrous, 99.5%), 4-dimethylamino pyridine (DMAP), proteinase K, *Pseudomonas* lipase, Porcine Pancreatic lipase, Pig Liver esterase, Protease, phosphate buffered saline and tris HCl buffer solution were purchased from Sigma Aldrich. Ethyl Acetate (EtOAc, analytical grade), tetrahydrofuran (THF, HPLC-grade), methanol (MeOH, analytical grade), Sodium hydroxide (1 M), magnesium sulfate, silica (TLC and free powder) and petroleum ether (Pet. Ether, analytical grade) were purchased from Fischer. Glycidyl Methacrylate (GLYMA, > 95%) was purchased from TCI chemicals. BHEMAG and BHEMAT were synthesised in the research laboratory. All materials were used as received unless otherwise stated.

### Enzyme catalysed hydrolysis and kinetics studies
The studied polymer samples (~ 30 mg) were placed in the prepared enzymatic media and kept at a constant temperature of 37 °C in a stirrer incubator at a rotation speed of 400 rpm. Enzymatic solutions were replaced at repeated time intervals depending on apparent product solubility to ensure maximum enzymatic activity. Sodium azide (2 mg) was added to each solution as an anti-mildew and anti-bacterial agent. Mass loss was measured after 1 month. The measurements were made by removal of the supernatant aqueous phase by pipette, followed by careful washing of the samples with deionised water and drying of the samples in a vacuum oven at

40 °C. Sample masses were measured with a balance displaying accuracy > 0.1 mg. Mass loss was calculated by division of the change in mass divided by the intial mass. To determine the influence of enzyme free fluid hydrolysis, polymer samples were subjected to the same temperature and spin in enzyme free buffer solution. PCL controls were utilised to demonstrate enzyme activity. For kinetic experiments, each polymer was separated into 10 individual vials and subjected to the same procedure as described above. When each time point was reached, the vial was submerged in ice to cease enzymatic activity.

### TBRT of multivinyl taxogens with a range of telogens
In a typical TBRT experiment, MVT (LDMA/BHEMAG/BHEMAT) (varying equiv.), telogen (DDT/CHT/TG) (1 equiv.), AIBN (1.5 mol% relative to vinyl bonds) and EtOAc (50 wt. %) were loaded into a 25 mL round-bottomed flask equipped with a magnetic stirrer bar. The solution was homogenised by agitation and a sample was extracted for $^1$H NMR spectroscopic analysis prior to initiation. The solution was deoxygenated whilst stirring for 20 min using a nitrogen purge. The solution was then heated to 70 °C with stirring and allowed to proceed for 24 h. The reaction was ceased by exposure to air and cooling to ambient temperature. A sample of the crude reaction mixture was extracted for $^1$H NMR spectroscopic analysis. The remaining sample was diluted with THF ( < 10 mL) to reduce the viscosity, and precipitated into cold methanol or Petroleum ether (depending on the telogen used). Precipitations typically afforded a white or clear precipitate and turbid supernatant. The precipitate was washed further with fresh antisolvent (3 × 50 mL) and subsequently dried *in vacuo* overnight at 40 °C. Finally, samples of the purified polymer were taken for $^1$H NMR and TD-SEC analysis.

### Synthesis of novel divinyl methacrylic MVTs by base catalysed esterification
Hydroxyethyl methacrylate (HEMA) (2.2 eq vol), DMAP (0.4 eq), triethylamine (2.2 eq) and DCM were charged to a suitably sized reaction flask under a nitrogen atmosphere. The reaction was cooled to 0 °C with stirring. A solution of glutaryl chloride or terepthaloyl chloride (1.0 eq) in DCM was charged to the stirred, cooled reaction mix at such a rate as to maintain the temperature below 5 °C. The resultant yellow solution was stirred out for 1 h at 0 °C then allowed to warm to room temperature overnight with stirring. The reaction was sampled to confirm COR. Suspended triethylamine hydrochloride was removed via vacuum filtration. The reaction mix was washed with 1 molar sodium carbonate (3 × 20 vol), 1 molar hydrochloric acid (3 × 20 vol) and sat. brine (1 × 20 vol). The DCM solution was subsequently dried (MgSO$_4$), filtered and the solvent removed under vacuum prior to purification by column chromatography (silica, 4:1 hexane/ethyl acetate) to afford the desired product after removal of solvent. Analysis

by [1]H NMR spectroscopy, [13]C NMR spectroscopy, elemental analysis and mass spectroscopy was then completed.

### Linear telomerisation of 2-hydroxethyl methacrylate (HEMA)

In a typical linear telomerisation, HEMA, (40.00 mmol, 2 equiv.), TG (20.00 mmol, 1 equiv.), AIBN (98.5 mg, 0.60 mmol) and EtOAc (90.80 mmol) were loaded into a 25 mL round-bottomed flask equipped with a magnetic stirrer bar. The solution was homogenised by agitation and a sample was extracted for [1]H NMR spectroscopic analysis of the reaction mixture prior to initiation to determine initial ratio. The solution was deoxygenated whilst stirring for 30 min using a nitrogen purge. The solution was then heated to 70 °C with stirring and allowed to proceed for 24 h. The reaction was ceased by exposure to air and cooling to ambient temperature. A sample of the crude reaction mixture was extracted for [1]H NMR spectroscopic analysis to determine monomer conversion. The crude samples were concentrated *in vacuo* initially using a spiral evaporator and finally a vacuum oven at 40 °C for 24 hours. Crude products were then analysed using MALDI-TOF mass spectroscopy

### Instrumentation

Nuclear Magnetic resonance (NMR) Spectroscopy: [1]H NMR & [13]C NMR spectra were recorded on a Bruker Advance DPX-400 MHz spectrometer. Samples were analysed in deuterated chloroform (CDCl₃) at ambient temperature. Chemical shifts (δ) are reported in parts per million (ppm) relative to the known solvent signal (δ = 7.26 ppm). Elemental analysis: CHNS elemental analyses were recorded on a ThermoFlash EA 1112 series elemental analyser using a Vario microcube. Triple-Detection Size Exclusion Chromatography (TD-SEC): All TD-SEC analysis of branched polymers was performed using a Malvern Viscotek instrument, equipped with a GPCmax VE2001 auto-sampler, two Viscotek T6000M columns (and a guard column) and a triple detector array TDA305 containing a refractive index (RI) detector VE3580 and a 270 Dual Detector (light scattering and viscometer). A mobile phase of THF containing 2 v/v % of triethylamine at 35 °C was used at a flow-rate of 1 mL/min. All samples were dissolved at 10 mg/mL in the eluent and filtered through a 0.2 μm PTFE syringe filter prior to injection (100 μL). Narrow and broad polystyrene standards were used to establish methods. All SEC analysis of linear telomers was performed using a Malvern Viscotek instrument, equipped with a GPCmax VE2001 auto-sampler, a mixed column setup of one T2000 column and one T1000 column in series (and a guard column) and a triple detector array TDA302 containing a refractive index (RI) detector VE3580 and a 270 Dual Detector (light scattering and viscometer). A mobile phase of THF at 35 °C was used at a flow-rate of 1 mL/min. All samples were dissolved at 10 mg/mL in the eluent and filtered through a 0.2 μm PTFE syringe filter prior to injection (100 μL). All SEC associated data were estimated using Omnisec 4.7 software. Mass Spectroscopy: Matrix-assisted laser desorption ionisation – time of flight (MALDI-TOF) mass spectra of linear telomers were analysed using a Bruker Autoflex Mass Spectrometer (Materials Innovation Factory, Liverpool, UK). Spectra for samples were each the sum of 500 shots acquired in positive-reflectron mode. Cesium triiodide (CsI₃) and α-cyano-4-hyrdroxycinnamic acid (HCCA) were used as the mass scale calibrant and matrix, respectively. Both the matrix and samples were prepared at 10 mg/mL in THF. The solutions were combined at a 5:1 v/v ratio of matrix to sample. 2 μL of the prepared solutions were deposited onto stainless-steel sample plates and air dried prior to analysis. Differential Scanning Calorimetry (DSC): Differential scanning calorimetry was performed on a TA instruments DSC 25 with an RSC 90 cooling system. The DSC pans used were standard aluminium with hermetic lids. Calibration was completed using an indium standard and the blank reference was an empty sealed aluminium pan with hermetic lid. Polymer samples were analysed by DSC using an empty pan as a reference standard. An identical method utilised for each polymer. A heat/cool/heat cycle was employed to remove the polymers' thermal history. Samples were initially heated from room temperature to 110 °C at a heating rate of 10 °C/min. A cooling rate of 10 °C/min was then employed to cool samples to −80 °C; this was followed by a 10 min isotherm.

Each polymer was then heated to 90 °C at a heating rate of 5 °C/min. The second heating profile following the isotherm was used for thermal analysis.

## Data availability

All data generated during this study supporting its findings are available within the manuscript and the Supplementary Information. All data is available from the corresponding author upon reasonable request.

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

## Acknowledgements
The Engineering & Physical Sciences Research Council (EPSRC) are grateful acknowledged for funding through grant EP/R010544/1. SM is grateful for studentship funding from EPSRC and Unilever. OBPL is grateful for studentship funding from EPSRC and the University of Liverpool. SRC is grateful for studentship funding from Itaconix. The authors would like to thank the Materials Innovation Factory (University of Liverpool)

## Author contributions
S.M. was responsible for conceptualisation, methodology, experimentation, investigation, data curation, formal analysis, visualisation and editing of the original draft. O.B.P-L., S.F., S.R.C., S.L. and S.W. all contributed to validation. C.F. and P.P. provided funding support and supervision. P.C. contributed to supervision, project administration, validation, and manuscript review. S.P.R. was responsible for funding acquisition, conceptualisation of the original research programme, methodology, validation, visualisation, supervision, project administration and manuscript review and editing,

## Competing interests
S.R.C., P.C. and S.P.R. are co-inventors on patents that protect the TBRT chemistry; these patents have been licensed to Scott Bader and form the basis of Polymer Mimetics Ltd (Company number 12598928). No other co-authors have any competing interests.
