## [Peer Review File · Communications Chemistry]

Controlling enzyme hydrolysis of branched polymers synthesised using transfer-dominated branching radical telomerisation via telogen and taxogen selection

This manuscript has been previously reviewed at another Nature Portfolio journal. This document only contains reviewer comments and rebuttal letters for versions considered at *Communications Chemistry*.Reviewers' comments:

Reviewer #1 (Remarks to the Author):

Rannard et al. described the relationship between monomer structure and enzymatic degradation of branched polymers prepared by TBRT, a synthetic method developed by the same authors. The authors have previously published several papers on TBRT, focusing on the synthesis of branched polymers. However, this study is the first to examine enzymatic degradability. Although the synthesis method is based on radical reactions, it employs a unique approach that can be classified as a polyaddition reaction. The effects of the main chain and side groups of branched polymers on enzymatic degradation were carefully examined. The description of the experiment is also accurate. Therefore, I recommend to accept this paper in Commun. Chem., although the following points need to be reconsidered before publication.

There is a lack of proper citations to support important claims. For example,

P3 'In many cases....into small molecules'

P3 'Branched polymers...non-degradable polymers'

P3 'extended backbones containing C-C bonds pose specific issues'

Regarding this claim, for example, the trimer of the vinyl monomer (C2) is only C6, and it is debatable how serious the issue the authors claim is. However, although the authors' claims may be exaggerated, they are not incorrect. Therefore, this statement was not problematic.

On the other hand, I feel it is not proper that no other approaches to prevent 'extended backbones containing C-C bonds' are cited. For example, alternating copolymerization is a popular strategy in this field. Some notable studies have reported the following:

ACS Macro Lett. 2024, 13, 368

Macromolecules 2018, 51, 14, 5079

Macromol. Rapid Commun. 1998, 19, 149

Angew. Chem. Int. Ed. 2023, 62, e202217365

P4

A general approach to telomerization, which is a strategy that actively performs chain transfer reactions to synthesize branched polymers, should be explained. The manuscript contains an explanation of telomerization defined 70 years ago but lacks an explanation of its subsequent history. Historically, branched polymers have been synthesized by the coexistence of a monovinyl compound and an allylic chain transfer agent capable of addition-fragmentation chain transfer. In contrast, TBRT combines a divinyl monomer and chain transfer agent. The difference between conventional approaches and TBRT is the position of the second vinyl group (in the monomer or in the chain transfer agent) and the residual rate of unsaturated bonds. TBRT is a term coined by the authors and used in the following papers: Polym. Chem., 2023, 14, 5102 ; Polym. Chem., 2023, 14, 1905; RSC Adv., 2022, 12, 31424. However, this is not an established concept, such as ATRP or RAFT. For this reason, it is preferable to write the explanation of TBRT ('Recently, a new synthetic strategy...was reported') in the active voice, such as 'We recently reported...'. As this is an interesting technique, please accurately portray the background and assert its originality.

Is Figure 1A unnecessary? Figure 1A is an abstraction of 1B; although it takes up more space, it does not contain as much information as Figure 1B.

It is necessary to introduce previous research by other groups on the correlation between branching structure and degradability, and compare it with the idea of this research. For example, in radical polymerization, the following research is known.

Polym. Chem., 2023, 14, 5154

Figure 2: There is no real meaning in arranging the six abstract figures. In my opinion, the combination of the schematic diagram in Figure 1A with the structural formulas of each taxogene and telogen is sufficient.

P14

'the p([TG-LDMA]75-stat-[TG-BHEMAG]25) showed almost double the mass loss observed in the

p(TG-LDMA) homopolymer'

Does this mean that the mass loss, which was approximately 10% for p(TG-LDMA), increased to approximately 20% for p([TG-LDMA]75-stat-[TG-BHEMAG]25). The mass reduction of p([TG-BHEMAG]) is approximately 80%. Therefore, if 25 mol% is [TG-BHEMAG], this result is expected. Unless the degradation efficiency is significantly increased compared to the introduction rate of [TG-BHEMAG], there is no need to state that it is 'almost double.' A description of only the mass reduction values was sufficient.

Figure 5A,B

Because the degradation is evaluated in terms of mass loss, the copolymer composition should be expressed in mass ratio rather than molar ratio. However, I am aware that these molar ratios are customarily used.

Reviewer #3 (Remarks to the Author):

I co-reviewed this manuscript with one of the reviewers who provided the listed reports. This is part of the Communications Chemistry initiative to facilitate training in peer review and to provide appropriate recognition for Early Career Researchers who co-review manuscripts.

Reviewer #4 (Remarks to the Author):

This manuscript reports initial results on the biodegradability of TBRT-prepared branched polyesters. The main results is that this biodegradability can be modulated by the choice of taxogen, but, importantly, also by the choice of telogen. The results are clearly presented and the conclusion drawn are sound and convincing. Adequate control experiments have been carried out and the scientific methods followed are flawless. Given the current urgent need for more biodegradable materials in applications where collection and recycling cannot be performed this TBRT-branched polyesters endowed with biodegradability are an important step in the right direction. This manuscript will undoubtedly be of high interest to polymer and materials scientists and engineers. I thus recommend publication.

Dear Dr Honda,

Manuscript ID: COMMSCHEM-24-0222-T

TITLE: Controlling enzyme hydrolysis of branched polymers synthesised using TBRT via telogen and taxogen selection.

Thank you for your recent help and communication regarding our submission to *Communications Chemistry*.

We thank the reviewers for their time and comments, and I provide a detailed response to each of their comments below.

Reviewer #1:

We note that Referee 1 has commented favourably on the novelty and rigour of our report and recommends publication after consideration of a number of points. These are considered and addressed below:

There is a lack of proper citations to support important claims. For example,

1) *P3 'In many cases....into small molecules'*

Response: This comment followed a previous sentence that contained references 4-8 and, in itself, is building on these cited works. I have now added “4-8” at the end of the sentence to make this clear

2) *P3 'Branched polymers...non-degradable polymers'*

Response: This comment also followed a previous statement that contained references 9-10. I have now added “9-10” at the end of the sentence to make this clear

3) *P3 'extended backbones containing C-C bonds pose specific issues'*
Regarding this claim, for example, the trimer of the vinyl monomer (C2) is only C6, and it is debatable how serious the issue the authors claim is. However, although the authors' claims may be exaggerated, they are not incorrect. Therefore, this statement was not problematic. On the other hand, I feel it is not proper that no other approaches to prevent 'extended backbones containing C-C bonds' are cited. For example, alternating copolymerization is a popular strategy in this field. Some notable studies have reported the following: ACS Macro Lett. 2024, 13, 368; Macromolecules 2018, 51, 14, 5079; Macromol. Rapid Commun. 1998, 19,149; Angew. Chem. Int. Ed. 2023, 62, e202217365

Response: We fully agree with the reviewer that it would be appropriate to make reference to alternating copolymerisations and free radical routes using cyclic ketene acetals. We have taken their recommendation to include ACS Macro Letters 2024,13, 368 as a new reference and also added Polym. Chem., 14, 5154-5165 (2023) that is mentioned later in this review. These are very recent examples of this approach. The following text has also been added (note new references 9 and 10)

“Alternating copolymerisation approaches have been utilised to introduce cleavable units into chain-growth polymers and avoid long segments derived from C-C bond formation.⁹ Additionally, homopolymerisation of monomers such as cyclic ketene acetals via radical ring-opening polymerisation generate polymers with repeating ester backbones.¹⁰”

- 4) *P4 A general approach to telomerization, which is a strategy that actively performs chain transfer reactions to synthesize branched polymers, should be explained. The manuscript contains an explanation of telomerization defined 70 years ago but lacks an explanation of its subsequent history. Historically, branched polymers have been synthesized by the coexistence of a monovinyl compound and an allylic chain transfer agent capable of addition-fragmentation chain transfer. In contrast, TBRT combines a divinyl monomer and chain transfer agent. The difference between conventional approaches and TBRT is the position of the second vinyl group (in the monomer or in the chain transfer agent) and the residual rate of unsaturated bonds. TBRT is a term coined by the authors and used in the following papers: Polym. Chem., 2023, 14, 5102 ; Polym. Chem., 2023,14, 1905; RSC Adv., 2022,12, 31424. However, this is not an established concept, such as ATRP or RAFT. For this reason, it is preferable to write the explanation of TBRT ('Recently, a new synthetic strategy...was reported') in the active voice, such as 'We recently reported...'. As this is an interesting technique, please accurately portray the background and assert its originality.*

Response: We are a little concerned about this comment from Reviewer 1 as we have published 10 papers using TBRT techniques, given a significant number of conference presentations (including 2 National ACS meetings), received awards, created a licenced joint venture with Scott Bader, and recently been informed of our 6th national grant within a portfolio of 6 patent families. We feel that actively re-introducing TBRT in each of our publications is duplicating the existing literature and have been criticised recently within reviews of other manuscripts for doing this. We do understand the sentiment, however, and do already commit two paragraphs, a figure, and 9 citations to explaining the concept of telomerisation, establishing the fact that the terminology we use is not new (as previously criticised by other reviewers of our initial reports), the main current focus of catalytic modification of unsaturated natural feedstocks, and explaining the difference between conventional telomerisation and TBRT. That said, we have now included a highly important review of telomerisation by Bernard Boutevin, one of the leading proponents of the use of telomerisation, as new reference 18, to cite the history of telomerisation for readers.

If this is not felt to be enough by the editor and the reviewers, we can include more but it would be a deviation from the main thrust of the manuscript, and we could be equally criticised for not explaining other concepts in more detail in the opening background sections of the report. We have modified the text to read in the active voice, as recommended by Reviewer 1; we usually refrain from this but again, we see the reviewer's point of view and have altered the text to read:

“We recently reported, a new synthetic strategy for high molecular weight branched polymers that employs conventional free radical chemistries to form materials with backbones that are analogous to step-growth polymers”

- 5) *Is Figure 1A unnecessary? Figure 1A is an abstraction of 1B; although it takes up more space, it does not contain as much information as Figure 1B.*

Response: We believe that this Figure is needed in its current form as it relates to the point made previously by this reviewer about ensuring the reader understands the work and has a clear view of the polymers being formed. We are careful to explain the concepts of TBRT in the two paragraphs but feel that this figure is highly informative to the reader that has not previously seen TBRT polymers.

- 6) *It is necessary to introduce previous research by other groups on the correlation between branching structure and degradability, and compare it with the idea of this research. For example, in radical polymerization, the following research is known. Polym. Chem., 2023, 14, 5154*

Response: We thank the reviewer for this comment and have included this specific reference (new reference 10) earlier in the manuscript and again within the following new text.

“To the best of our knowledge, there are no current commercial polymers resulting from radical ring opening polymerisation of cyclic ketene acetals;¹¹ however, recent studies have shown that relatively light branching within these materials, resulting from uncontrolled back-biting reactions and initiation from the polymer backbone, may impede hydrolysis and biodegradation.^{10”}

Although we have included this work, it is clear that studying the impact of uncontrolled branching in an otherwise linear polymerisation does not fully relate to the work here where we are introducing susceptibility to enzyme hydrolysis into our targeted highly branched polymers. The reviewer is correct though that the context is useful and helps us to differentiate our report given the increase in enzymatic hydrolysis that we see, rather than the restriction of hydrolysis and biodegradation reported with these earlier studies.

- 7) *Figure 2: There is no real meaning in arranging the six abstract figures. In my opinion, the combination of the schematic diagram in Figure 1A with the structural formulas of each taxogene and telogen is sufficient.*

Response: We respectfully disagree with the reviewer. As this reviewer points out in their earlier comments, readers are not fully conversant with TBRT structures. We believe that the more we can present visual aids to allow them to understand the structures and the variations that we are designing within the materials, along with the nominal repeating structures, the better the reader will understand the thrust of the manuscript. This has not been raised by other reviewers and we have not changed this aspect of the paper.

- 8) *P14 ‘the p([TG-LDMA]75-stat-[TG-BHEMAG]25) showed almost double the mass loss observed in the p(TG-LDMA) homopolymer’ Does this mean that the mass loss, which was approximately 10% for p(TG-LDMA), increased to approximately 20% for p([TG-LDMA]75-stat-[TG-BHEMAG]25). The mass reduction of p([TG-BHEMAG]) is approximately 80%. Therefore, if 25 mol% is [TG-BHEMAG], this result is expected. Unless the degradation efficiency is significantly increased compared to the introduction rate of [TG-BHEMAG], there is no need to state that it is ‘almost double.’ A description of only the mass reduction values was sufficient.*

Response: We respectfully disagree with the reviewer. Clear communication of the results and carefully explaining the data to the reader is key to generating understanding of the effects of the polymer modifications; this may require explanation from different perspectives. We do not agree that anything is to be “expected” within these materials as they are novel and previously untested under such conditions. As this reviewer points out in previous comments, especially with respect to the previous reports of branched polymer degradation, it is not always obvious which structural differences will impact the behaviour of the polymers or to what extent. Within ([TG-LDMA]75-stat-[TG-BHEMAG]25), for example, 75% of the branched architecture is not highly susceptible to enzymatic hydrolysis and the inclusion of TG-BHEMAG may have been ‘lost’ within the majority of the complex

macromolecules, resulting in no real impact at all. We have not modified the manuscript with respect to this comment.

9) *Figure 5A,B - Because the degradation is evaluated in terms of mass loss, the copolymer composition should be expressed in mass ratio rather than molar ratio. However, I am aware that these molar ratios are customarily used.*

Response: We thank the reviewer for this comment and fully understand the sentiment. In order to not confuse the reader and given the naming of the polymer structures is already established within the manuscript at this stage, it is felt that introducing another approach to referring to the polymers would lead to a lack of clarity and have accepted the reviewer's point that such naming is "customarily used".

That said, the reviewer has highlighted this figure and this has led to us identifying a mistake within the labelling of Fig 5C; p(TG-LDMA) and p(TG-BHEMAG) are incorrectly labelled in the legend and this has been addressed in our resubmission.

Reviewer #3:

I co-reviewed this manuscript with one of the reviewers who provided the listed reports. This is part of the Communications Chemistry initiative to facilitate training in peer review and to provide appropriate recognition for Early Career Researchers who co-review manuscripts.

Response: We thank the efforts of this co-reviewer and have not made any changes in respect of this comment.

Reviewer #4:

This manuscript reports initial results on the biodegradability of TBRT-prepared branched polyesters. The main results is that this biodegradability can be modulated by the choice of taxogen, but, importantly, also by the choice of telogen. The results are clearly presented and the conclusion drawn are sound and convincing. Adequate control experiments have been carried out and the scientific methods followed are flawless. Given the current urgent need for more biodegradable materials in applications where collection and recycling cannot be performed this TBRT-branched polyesters endowed with biodegradability are an important step in the right direction. This manuscript will undoubtedly be of high interest to polymer and materials scientists and engineers. I thus recommend publication.

Response: We thank this reviewer for their overwhelmingly positive comments and recommendation of publication

REVIEWERS' COMMENTS:

Reviewer #1 (Remarks to the Author):

The authors have seriously considered my comments and are revising the manuscript. While they disagree with some of them, they have responded with clear reasons and careful explanations. All of the revisions are reasonable. I recommend publishing this manuscript in this form.

Reviewer #3:

I co-reviewed this manuscript with one of the reviewers who provided the listed reports. This is part of the Communications Chemistry initiative to facilitate training in peer review and to provide appropriate recognition for Early Career Researchers who co-review manuscripts.